# Down-Regulation of miR-194-5p for Predicting Metastasis in Breast Cancer Cells

**DOI:** 10.3390/ijms23010325

**Published:** 2021-12-28

**Authors:** Yu-Ting Yen, Jou-Chun Yang, Jiun-Bo Chang, Shih-Chang Tsai

**Affiliations:** 1Department of Medical Research, China Medical University Hospital, China Medical University, Taichung 404332, Taiwan; d92449001@ntu.edu.tw; 2Drug Development Center, Institute of New Drug Development, Institute of Biomedical Sciences, China Medical University, Taichung 404332, Taiwan; 3Department of Biological Science and Technology, China Medical University, No. 100, Sec. 1, Jing-Mao Road, Taichung 406040, Taiwan; tsubasa131047@gmail.com (J.-C.Y.); billy8684@gmail.com (J.-B.C.)

**Keywords:** miR-194-5p, triple-negative breast cancer (TNBC), The Cancer Genome Atlas (TCGA), epithelial–mesenchymal transition (EMT), ZEB-1

## Abstract

MicroRNAs (miRNAs), as key negative regulators of gene expression, are closely related to tumor occurrence and progression. miR-194-5p (miR-194-1) has been shown to play a regulatory role in various cancers however, its biological function and mechanism of action in breast cancer have not yet been well explored. In this study, we use the UALCAN and LinkedOmics databases to analyze transcription expression in The Cancer Genome Atlas Breast Invasive Carcinoma (TCGA-BRCA). The epithelial-mesenchymal transition status of breast cancer cells was evaluated by wound-healing assay, trans-well assays, and gelatin zymography, while protein expression was assessed by Western blotting. miR-194-5p expression was found to be up-regulated in breast cancer clinical specimens but down-regulated in the triple-negative breast cancer (TNBC) cell line MDA-MB-231 and breast cancer clinical specimens in The Cancer Genome Atlas (TCGA). miR-194-5p significantly inhibited the expression of the epithelial marker ZO-1 and increased the expression of mesenchymal markers, including ZEB-1 and vimentin, in MDA-MB-231 cells. miR-194-5p significantly reduced the gelatin-degrading activity of matrix metalloproteinase-2 (MMP-2) and MMP-9 in zymography assays. In MDA-MB-231 cells and TCGA patient samples, ZEB-1 expression was significantly inversely correlated with miR-194-5p expression. High levels of miR-194-5p were associated with good overall survival. miR-194-5p regulates epithelial–mesenchymal transition (EMT) in TNBC. Our findings suggest that miR-194-5p functions as a tumor biomarker in breast cancer, providing new insights for the study of breast cancer development and metastasis.

## 1. Introduction

Breast cancer is a leading cause of death and the most commonly diagnosed cancer in women worldwide. Each year, more than 250,000 women are diagnosed with breast cancer, accounting for 30% of all female cancers [1]. In breast cancer, the cells are classified as primary or metastatic. Primary breast cancer cells expressing estrogen receptors (ER) or progesterone receptors (PR) can be treated with hormone therapy. Over 90% of cancer-related deaths are caused by metastases [2]. Metastases contribute to multiple complex processes, including epithelial–mesenchymal transition (EMT), anoikis resistance, and angiogenesis [3,4]. Triple-negative breast cancer (TNBC) is characterized by an absence of the estrogen receptor (ER), the progesterone receptor (PR), and HER-2 (human epidermal growth factor receptor 2). The recurrence and mortality rates associated with TNBC are high [5]. Due to the lack of these three receptors, endocrine therapy and targeted therapy are not effective against TNBC [6]. There is often a poor prognosis and drug resistance with chemotherapy [7]. Therefore, at present, it is urgent to design new drugs and develop effective therapeutic approaches for the treatment of TNBC [8]. By developing potential prognostic markers, we can better understand the molecular mechanisms that minimize tumor development [9].

The roles of miRNAs have been increasingly associated with breast cancer metastasis. miRNAs are short (about 21–25 nucleotides in length) non-coding RNA molecules that negatively control protein synthesis by specifically binding to the 3′-UTRs (3′-untranslated regions) of their target mRNAs [10]. miR-206, miR-221/222, and miR-34 have been shown to have inhibitory effects on breast cancer tumorigenesis and proliferation [11]. MiR-194-5p inhibits cell proliferation and migration in non-small cell lung cancer [12], deregulates acute myeloid leukemia [13], and inhibits tumor growth in colorectal cancer [14]. In addition, over-expression of miR-194-5p has been related to the recurrence of breast cancer [15] however, the exact cellular functions and molecular mechanisms of miR-194-5p remain to be investigated in breast cancer.

The metastasis and spread of cancer cells are the main reasons for the recurrence and death of cancer patients, as well as increased difficulty of treatment. The metastasis and spread of cancer cells are often accompanied by many physiological changes in cancer cells, which cause aggressive properties. The main phenomenon relating to metastasis is called EMT [16]. ZEB1 is an EMT transcription factor that facilitates metastasis in breast and pancreatic carcinomas [17,18]. Transforming growth factor-β (TGF-β) induces ZEB1 and promotes bone-specific metastasis of breast carcinomas [19,20]. ZEB1 expression exhibits a repressive role and contributes to anti-cancer drug resistance [21,22].

In the current study, the expression level of miR-194-5p is examined in breast cancer cell lines and clinical tissues. The potential miR-194-5p-directed target genes are identified and characterized, and the action mechanism of miR-194-5p in breast cell migration is investigated.

## 2. Results

### 2.1. miR-194-5p Expression in TCGA-BRCA

To examine the expression levels of miR-194-5p in different types of cancer, we used the TCGA and Genotype-Tissue Expression (GTEx) miRNA databases to perform systematic pan-cancer analysis. The results revealed that miR-194-5p is over-expressed in many cancer types, including READ, COAD, STAD, PAAD, BRCA, BLCA, PRAD, and HNSC (Figure 1A). A forest plot displaying the number of tumor and normal samples, the area under the ROC curve (AUC), and 95% confidence interval (CI) of the AUC for each cancer type in TCGA was used to visualize the results of the pan-cancer ROC analysis by CancerMIRNome bioinformatic analysis (Figure 1B) [23].

### 2.2. Low Expression of miR-194-5p in the Basal-like Group in Breast Cancer

The TCGA breast cancer cohort and corresponding clinical data (survival time, TMN stages, tumor site, and different clinicopathological characteristics) were downloaded from the publicly available TCGA database. CancerMIRNome (http://bioinfo.jialab-ucr.org/CancerMIRNome/, accessed on 15 August 2021) was used to examine the expression profile between miR-194-5p expression levels and different cancer stages. Our findings revealed that miR-194-5p expression was lower in TNBC than in the luminal groups of breast cancers (*p* = 0.036; Figure 2A).

### 2.3. Survival Analysis

To predict and discriminate the miRNA-based signature profiles, we constructed the Kaplan–Meier curve (K–M curve) to compare overall survival (OS) in different risk groups. The miR-194-5p levels showed a significant difference in the ROC curve (AUC ROC) of 0.7 (95% CI: 0.66–0.75) between breast cancer and adjacent normal tissues. The K–M curve suggested that breast cancer patients with high miR-194-5p expression levels had significantly higher overall survival (Figure 2B). Once aberrantly expressed miRNAs were identified, we wanted to further verify miRNA-194-5p for the development of breast cancer metastases in a mouse model [24]. Thus, miRNA RNA sequence analysis was performed, followed by analysis of the number of miRNAs whose expression was altered, in comparison with the corresponding time point control. Down-regulation of miR-194-5p was observed in the study at Days 3 and 10, post-injection of 4T1 cells (Figure 2C).

### 2.4. Functional Enrichment Analysis of miR-194-5p in Breast Cancer Patients

To further clarify the biological meaning behind miR-194-5p and its associated genes, we performed functional enrichment analysis using the LinkedOmics database. Differentially expressed genes correlated with miR-194-5p in breast cancer are presented as volcano plots in Figure 3A. The top 50 genes significantly positively correlated with miR-194-5p, chosen from LinkedOmics, are shown in Appendix A, and the top 48 genes significantly negatively correlated with miR-194-5p chosen from LinkedOmics are shown in Appendix A. We further performed GSEA to enrich and identify signaling pathway structures in which target genes correlated with miR-194-5p are involved. As shown in Figure 3B, major signaling pathway structures were focal adhesion, extracellular matrix (ECM)–receptor interaction, proteoglycans in cancer, EGFR tyrosine kinase inhibitor resistance, PI3K-Akt signaling pathway, and regulation of actin cytoskeleton. GSEA revealed negative enrichment of miR-194-5p-altered genes in two top gene sets: Focal adhesion and ECM–receptor interaction (Figure 3C).

### 2.5. miR-194-5p Inhibits Metastasis of Breast Cancer

RT-qPCR was used to examine the expression of miR-194-5p in breast cancer cell lines MCF-7 and MDA-MB-231, with low and high metastatic potential, respectively. The results in Figure 4A demonstrate that the low expression of miR-194-5p in MDA-MB-231 cells was 0.2-fold higher than that in MCF-7 cells (Figure 4A). Cell migration ability is an indicator of the metastatic potential of cancer cells. To evaluate the effect of miR-194-5p on cell migration, we examined the migration ability of MDA-MB-231 by wound-healing assay and Trans-well assay after treatment with miR-194-5p mimics or control miRNA (miR-ZIP). In the wound-healing assay, confluent transfected cells were scratched and observed for 24 h (Figure 4B). Treatment with miR-194-5p mimics led to a significant decrease in wound healing, compared to treatment with miR-ZIP in MDA-MB-231 cells. Similarly, the migration of transfected miR-194-5p mimic MDA-MB-231 cells was blocked in the Trans-well assay (Figure 4C). EMT is characterized by loss of polarity and epithelial markers. E-cadherin and ZO-1 are epithelial markers, whereas vimentin, α-SMA, and N-cadherin are mesenchymal markers, which define the epithelial or mesenchymal phenotype, respectively. However, we and other labs have found E-cadherin to be undetectable in MDA-MB-231 cells [25]. To further examine whether miR-194-5p inhibits EMT, breast cancer cell line MDA-MB-231 was used to observe the variations in ZO-1, ZEB1, and vimentin. ZO-1 was significantly down-regulated by miR-194-5p, while ZEB-1 and vimentin were significantly up-regulated (Figure 4D).

### 2.6. miRNA-194-5p Suppresses MMP2 and MMP9 Enzyme Activities

To determine whether miR-194-5p affected the gelatinolytic activities of MMP-2 and MMP-9, MDA-MB-231 cells were transfected with miR-194-5p mimics. Subsequently, the cell medium was collected and analyzed by gelatin zymography. MMP2 and MMP-9 enzyme activities in miR-194-5p-transfected MDA-MB-231 cells were lower than those in control miR-ZIP-transfected cells. The band at 72 kDa, representing MMP2 (Figure 5A), and the band at 95 kDa, representing MMP9 (Figure 5B), were reduced. These results indicate that miR-194-5p negatively regulated MMP2 and MMP9 enzyme activities.

### 2.7. miR-194-5p Directly Targeted FOXA1, BMI1, and ZEB1

miRNAs bind to the 3′-UTRs of their target genes and block protein synthesis. To identify the potential targets of miR-194-5p in breast cancer, the publicly available miRBase and TargetScan were used to screen potential targets. The 3′-UTRs of human FOXA1, Bmi1, and ZEB1 contained putative miR-194-5p-complementary sites, which bind to the seed region of miR-194-5p (Figure 6A). To validate whether FOXA1, Bmi1, and ZEB1 are potential targets of miR-194-5p, 3′-UTR fragments of human FOXA1, Bmi1, and ZEB1 containing wild-type miR-194-5p-complementary sequences were sub-cloned into the 3′-UTR of the firefly luciferase reporter gene vector (pMIR-REPORT). When miR-194-5p mimics were co-transfected with pMIR-REPORT containing wild-type FOXA1, Bmi1, and ZEB1 3′-UTRs, the firefly luciferase activities were obviously suppressed. This result indicated that miR-194-5p may down-regulate the expression of the FOXA1, Bmi1, and ZEB1 genes through binding miR-194-5p-complementary sequences at their 3′-UTRs (Figure 6B). The protein levels of FOXA1, Bmi1, and ZEB1 were assessed by Western blotting, in order to detect the inhibitory effect of miR-194-5p in MDA-MB-231 cells. There were significant negative correlations between miR-194-5p and the expression of FOXA1, Bmi1, and ZEB1 proteins in MDA-MB-231 cells (Figure 6C).

### 2.8. Expression of ZEB1, VIM, and MMP-2 Is Significantly Inversely Correlated with miR-194-5p Levels in TCGA-BRCA

We characterized the mRNA levels of ZEB1, VIM, BMI1, and MMP-2 in TCGA-BRCA. The data showed reverse correlations between miR-194-5p expression and ZEB1, VIM, BMI1, and MMP-2 expression in TCGA-BRCA (Figure 6D). These data further suggest that the up-regulation of ZEB1, VIM, BMI1, and MMP-2 in TCGA-BRCA may partly be due to the down-regulation of miR-194-5p. To identify major protein–protein interactions (PPIs) during EMT induced by ZEB1, we selected proteins that participate in EMT induced by ZEB1 and used STRING to acquire a PPI network (Figure 6E).

Figure 7 shows how miR-194-5p regulates cell migration by down-regulating the epithelial marker ZO-1 and up-regulating the mesenchymal markers ZEB1 and vimentin in MDA-MB-231 cells. 

## 3. Discussion

### 3.1. The Biological Functions of miR-194-5p in Breast Cancer Carcinogenesis

Although there are several suitable treatments for different types of breast cancer, breast cancer patients may develop recurrence. Most breast cancer patients die from the transfer of breast cancer cells to other tissues or organs, causing other complications and death. Most of these deaths are due to cancer metastasis, and the factors that trigger cancer metastasis remain unclear. MicroRNA research not only may reveal the mechanisms underlying metastasis, but can also suggest important implications for cancer diagnosis, prognosis, and treatment. miRNAs have been proposed as an alternative therapeutic approach for treating cancer in a clinical context [26]. In recent years, studies have shown that microRNAs regulate EMT-related proteins by up- or down-regulating their target genes [27]. The most commonly studied microRNA related to EMT is the miR-200 family, which plays an important role in EMT suppression, mainly by targeting ZEB [28].

Studies have revealed that miR-194-5p inhibits inflammation [29], and is involved in cartilage formation [30] and neuronal differentiation [31]. miR-194-5p is highly expressed in the intestine and liver [32,33]. During intestinal epithelial cell differentiation, miR-194-5p is induced by the hepatocyte nuclear factor [34]. The invasiveness and metastasis of interstitial liver cancer cells have been shown to be inhibited by miR-194-5p [35]. An association exists between a low expression of miR-194-5p and advanced gastric cancer [36]. miR-194-5p inhibits epithelial and mesenchymal metastasis of gastric cancer by targeting FOXM1 [22]. However, Cai et al. have shown that miR-194-5p promotes EMT in human colorectal cancer [37], while Yang et al. showed that a knockdown of miR-194-5p suppressed EMT in breast cell lines in an animal model [38]. Collectively, miR-194-5p acts as either an oncogene or a tumor suppressor gene in different cancers.

In the present study, we investigated the biological functions of miR-194-5p in breast cancer carcinogenesis. miR-194-5p is highly expressed in breast, intestinal, and liver cancers (Figure 1). Our findings suggest that the elevated expression of miR-194-5p is related to high overall survival (Figure 2C). Intriguingly, the relative expression of miR-194-5p was lower in the metastatic triple-negative breast cancer cell line (MDA-MB-231) than in the non-metastatic breast cancer cell line (MCF7) (see Figure 4A).

### 3.2. miR-194-5p Inhibits Invasion and Metastasis

We performed GSEA, and the results revealed that miR-194-5p is involved in ECM-receptor interactions (Figure 3B). These findings are consistent with those of previous studies. In endometrial cancer cells, miR-194-5p inhibits epithelial–mesenchymal transition by targeting Bmi-1 [39]. miR-194-5p inhibits the invasion and metastasis of interstitial hepatocellular carcinoma cells [35].

The results of wound-healing and Trans-well assays revealed that the migration of MDA-MB-231 cells was inhibited by miR-194-5p (Figure 4B,C). miR-194-5p increased the expression of an epithelial marker (ZO-1) and decreased that of a mesenchymal marker (vimentin) by Western blot analysis (Figure 4D). According to the zymography data, miR-194-5p inhibited the gelatin-degrading activities of MMP-2 (Figure 5A) and MMP-9 (Figure 5B). The TargetScan program was used to search for and verify miR-194-5p target genes, including AKT2, Bmi-1, FOXA1, and SOX5 (Figure 6A,B). The protein expression levels of miR-194-5p target genes were decreased after transfecting the miR-194-5p precursor into breast cancer cells (Figure 6C). Our results prove that miR-194-5p can inhibit EMT in breast cancer. In metastatic TNBC, miR-194-5p could serve as a useful biomarker. Using miR-194-5p to regulate TNBC migration is a novel strategy for preventing the migration of the disease.

## 4. Materials and Methods

### 4.1. Bioinformatic Analysis

The TCGA database contains high-throughput sequencing data and prognostic data. LinkedOmics was used to analyze omics data in cancer. We measured the receiver operating characteristic (ROC) curve for overall survival between risk patients, stratified by CancerMIRNome (http://bioinfo.jialab-ucr.org/CancerMIRNome/, accessed on 15 August 2021).

### 4.2. MiRactDB

We analyzed the miRNA–gene expression profiles between normal and cancer tissues using miRactDB (https://ccsm.uth.edu/miRactDB/, accessed on 15 August 2021) [40], a database for characterizing miRNA–gene interactions.

### 4.3. UALCAN Analysis

UALCAN is an open and comprehensive web portal [41], which is used for in-depth analyses of RNA-seq level 3 TCGA data. UALCAN was applied to study the relative expression levels of genes across normal and tumor tissues, as well as to characterize multiple cancer sub-types, such as TMN stages, tumor location, tumor grade, and different clinicopathological characteristics. In this study, the UALCAN database was applied to analyze the prognosis of miR-194-5p in breast cancer.

### 4.4. LinkedOmics Analysis

Thirty-two TCGA cancer-associated multi-dimensional data sets were selected and analyzed using the LinkedOmics database (visited at http://www.linkedomics.org/login.php/, accessed on 15 August 2021) [42]. The differentially expressed genes related to miR-194-5p were selected from the breast cancer cohort of the TGCA database through the LinkFinder module. The Pearson correlation coefficient was used to evaluate the correlation of results. The LinkInterpreter module assayed the pathways and networks across differentially expressed genes related to miR-194-5p. Gene set enrichment analysis (GSEA) and Kyoto Encyclopedia of Genes and Genomes (KEGG) pathway analysis are two approaches for interpreting gene expression profiles based on the three classes of biological processes, molecular functions, and cellular components. The pathway and network analyses were ranked through GSEA and KEGG. A cut-off *p*-value was set at 0.05 as the rank standard.

### 4.5. Analyses of Interactive Network and Modules

STRING (accessed through http://string-db.org/, accessed on 15 August 2021) is a known database used to seek protein–protein interactions. A protein–protein interactive network was established by screening out co-expressed genes with correlation scores above 0.4 [43].

### 4.6. Cell Cultures

MCF7 and MDA-MB-231 human breast adenocarcinoma cell lines were chosen for a comparison of migration in this study. Two cell lines were originally obtained from ATCC and maintained in DMEM/F-12 supplemented with 10% fetal bovine serum (FBS), 100 U/mL of penicillin, and 100 μg/mL of streptomycin under 5% CO_2_ in a 95% humidified incubator at a temperature of 37 °C.

### 4.7. Isolation of RNA Samples and Quantification of MicroRNA Expression

TRIzol Reagent (Cat. No. 15596026, Invitrogen, Carlsbad, CA, USA) was used to extract RNA samples from cells, according to the manufacturer’s instructions and as described previously [44]. RT-qPCR (reverse transcription-quantitative polymerase chain reaction) assays were performed for detecting miR-194-5p using TaqMan miRNA assays (Cat. No. 4427975, Applied Biosystems, Foster City, CA, USA), where U6 was the reference gene control. The RT settings and PCR cycling conditions were chosen according to the manufacturer’s instructions. The threshold cycle (Ct) values were obtained from triplicate RT-qPCR assays and analyzed using QuantStudio 3 software (Life Technologies, Carlsbad, CA, USA).

### 4.8. Predicting miR-194-5p Target Genes and Constructs

The web-based miRNA target prediction software miRBase (http://mirbase.org/, accessed on 15 August 2021) and TargetScan (www.targetscan.org/, accessed on 15 August 2021) were used to predict binding between the miR-194-5p seed sequence (about 6–8 nucleotides in length) and the 3′-untranslated region (3′-UTR) sequence of the target genes. The miR-194-5p sequence and the 3′-UTR sequence (about 21–23 nucleotides in length) of the potential target genes were cloned into a SpeI/HindIII-digested pMIR-REPORT vector (Applied Biosystems, Foster City, CA, USA), as described previously [44]. The following primer sequences were used for cloning miR-194-5p and miR-194-5p target genes: miR-194-5p forward, 5′-CTAGTTCCACATGGAGTTGCTGTTACAGGATCCA-3′ and reverse, 5′-AGCTTGGATCCTGTAACAGCAACTCCATGTGGAA-3′; FOXA1 forward, 5′-CTAGTCCCCAGTGCAAAAGACTGTTACTGGATCCA-3′ and reverse, 5′-AGCTTGGATCCAGTAACAGTCTTTTGCACTGGGGA-3′; Bmi1 forward, 5′-CTAGTTTTACATATATTGCTGTTACTGGATCCA-3′ and reverse, 5′-AGCTTGGATCCAGTAACAGCAATATATGTAAAA-3′; and ZEB1 forward, 5′-CTAGTCATTTTTAAGTTCCTTGTTACATGGATCCA-3′ and reverse, 5′- AGCTTGGATCCTGTAACAGCAACTCCATGTGGAA-3′.

### 4.9. Transfection of miR-194-5p into MDA-MB-231 Cells

MDA-MB-231 cells were inoculated in 6-well plates 18 to 24 h prior to transfection and reached 70–80% confluency. The cells were transfected with 2 µg of the hsa-miR-194-5p precursor plasmid (Cat# PMIRH1942AA-1) purchased from System Biosciences (Palo Alto, CA, USA), or 2 µg of the plasmid vector (as a control) using OMNIfect™ transfection reagent (cat no. OTR1004, Transomic, Huntsville, AL, USA), according to the manufacturer’s instructions.

### 4.10. Luciferase Reporter Gene Assays

Luciferase reporter gene enzyme activity was detected using a Dual-Luciferase^®^ Reporter Assay System (Cat. No. E1910, Promega, Madison, WI, USA). MDA-MB-231 cells were co-transfected with 500 ng of the hsa-miR-194-5p precursor plasmid, 1.0 μg of a pMIR-REPORT construct containing the 3′-UTR sequence of the potential target gene, and 500 ng of pRL-CMV for 48 h. Relative luciferase activity was determined by calculating the ratio of firefly-to-Renilla luciferase activity. Three independent experiments were performed.

### 4.11. Wound-Healing Assay

MDA-MB-231 cells transfected with a miR-194-5p precursor plasmid or an empty vector were seeded into ibidi culture inserts (Cat. No. 81176, ibidi GmbH, Gräfelfing, Munich, Germany) for wound-healing assay, according to the manufacturer’s instructions. The inserts were gently removed and the attached cells were maintained in 2-mL DMEM with 1% FBS for 24 h. Images were captured at 0 and 24 h. The gap distance in three fields was measured, in order to calculate the percentage of cell migration and were photographed under an inverted microscope (Leica, Wetzlar, Hessen, Germany) at a magnification of 100×. The gap distance was quantitatively evaluated using ImageJ software version 1.41 (National Institutes of Health, Bethesda, MD, USA).

### 4.12. Trans-Well Migration Assays

To detect cell migration, chambers with an 8-µm pore size (Transwell, Cat. No. 3464, Costar) were chosen and placed into 24-well culture plates, as described previously [45]. Briefly, miR-194-5p or an empty vector were transfected into MDA-MB-231 cells for 48 h and trypsinized. Approximately 5 × 10^4^ cells were inoculated in the upper chamber, supplemented with a 400-µL DMEM/F12 medium containing 10% FBS, which was added in the lower chamber for 24 h. Migrated cells were fixed with cold methanol and stained with 0.5% crystal violet for 1 h at room temperature. Purple migrated cells were photographed under a Leica light microscope (Leica, Wetzlar, Hessen, Germany) at 100× magnification. The number of migrated cells was counted. 

### 4.13. Gelatin Zymography

MDA-MB-231 cells were transfected with miR-194-5p or empty vector for 24 h. Transfected cells (8 × 10^5^ cells/well) were seeded into 24-well plates for another 24 h and cultured in a serum-free DMEM/F-12 medium for 24 h at 37 °C. Media were collected and analyzed by running gels to determine MMP-2 enzyme activity through gelatin zymography, as previously described [45].

### 4.14. Western Blot Analysis

Protein expression levels were assessed using Western blot analysis, as previously described [44]. Briefly, equal amounts of protein (100 µg) were loaded into wells and separated using a 10% sodium dodecyl sulfate-polyacrylamide gel. Protein samples were transferred onto polyvinylidene difluoride membranes (EMD Millipore, Billerica, MA, USA). The membranes were blocked in PBS-T (PBS containing 0.1% Tween-20) with 5% non-fat milk, then incubated with the indicated primary antibodies overnight at 4 °C. Primary antibodies were probed against: ZO-1 (Cat. NO. GTX108613; 1:1000 dilution), ZEB1 (Cat. NO. GTX55847; 1:1000 dilution), Vimentin (Cat. NO. GTX112661; 1:1000 dilution), FOXA1 (Cat. NO. sc-514695; 1:1000 dilution), Bmi1 (Cat. NO. GTX114008; 1:1000 dilution), and GAPDH (Cat. NO. MAB374; 1:5000 dilution). Subsequently, horseradish peroxidase-conjugated secondary antibody was added into the membranes and incubated for 1 h at room temperature. Protein bands were visualized using an enhanced chemiluminescence detection kit (GE Healthcare Bioscience, Piscataway, NJ, USA). The intensities of sample bands were quantified using ImageJ software version 1.41 (NIH). Quantitative expression levels of proteins were normalized to that of GAPDH.

### 4.15. Super-Enhancer (SE) Analysis

H3K27ac ChIP-seq data were downloaded from 15 human tissues, including 64 samples from ENCODE [46]. ROSE (Rank Ordering of Super-enhancers) is a tool for investigating SE profiles, and the H3K27ac signal surrounding a miR-194-5p site was analyzed [47].

### 4.16. Statistical Analysis

All experiments were performed independently at least three times. Statistical comparisons were performed using the GraphPad Prism 7 software. One-way ANOVA and Tukey’s honest significant difference test were used for multiple comparisons, while an unpaired Student’s *t*-test was used to compare the mean of two independent groups. *p* < 0.05 was considered to denote statistical significance.

## 5. Conclusions

Our results suggested that miR-194-5p is down-regulated in TNBC. miR-194-5p inhibited the migration ability of metastatic MDA-MB-231 cells. We demonstrated that the protein expression of the EMT-associated transcription factor Zeb-1 and mesenchymal marker vimentin was decreased, while that of the epithelial marker ZO-1 was increased in miR-194-5p mimic-transfected MDA-MB-231 cells. In addition, the enzyme activities of MMP-2 and MMP-9 were suppressed by miR-194-5p in miR-194-5p mimic-transfected MDA-MB-231 cells. Therefore, miR-194-5p shows potential as a useful biomarker in metastatic TNBC.

## Figures and Tables

**Figure 1 ijms-23-00325-f001:**
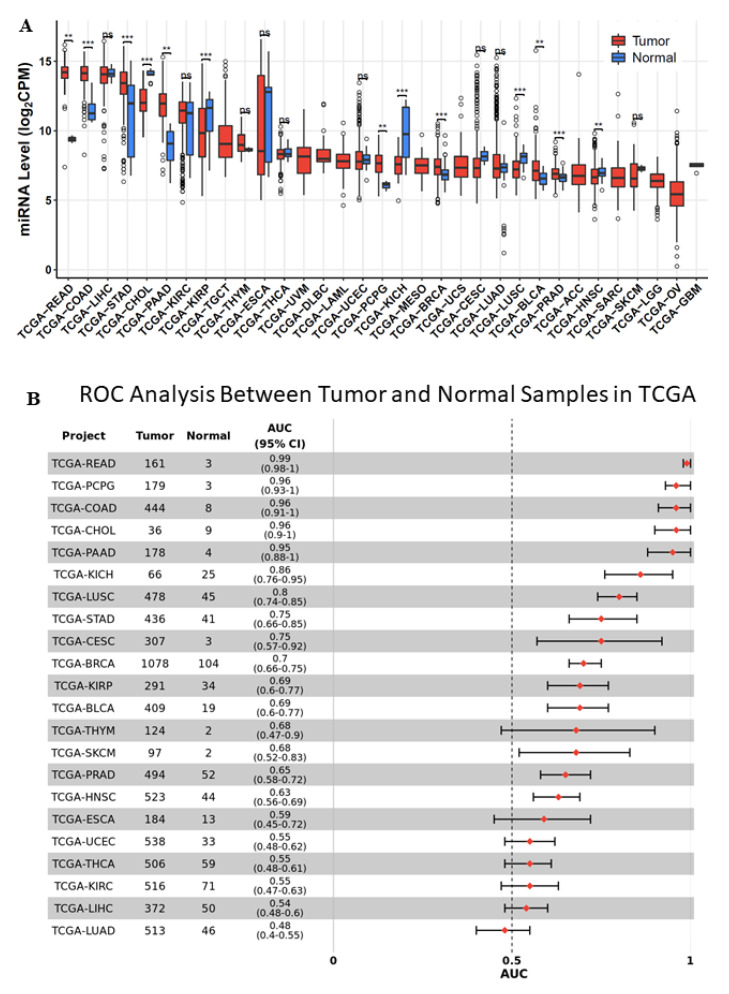
miR-194-5p expression in The Cancer Genome Atlas (TCGA) pan-cancer: (**A**) Pan-cancer expression of miR-194-5p in TCGA. Data are presented as “− delta CT” using box and whisker plots. **, *p* < 0.005; ***, *p* < 0.0005; n.s., not significant. (**B**) Forest plot visualizing pan-cancer survival analysis across all TCGA projects.

**Figure 2 ijms-23-00325-f002:**
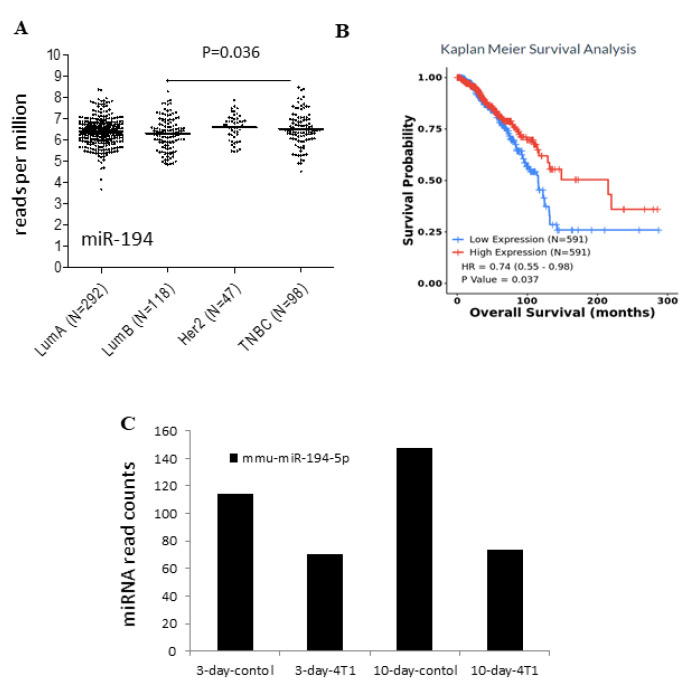
Association of miR-194-5p expression with clinicopathologic characteristics: (**A**) miRNA expression was analyzed in PAM50-defined breast tumor sub-types from TCGA. Gene expression differences were plotted across the triple-negative breast cancer (TNBC), HER2-positive (Her2), luminal A (LumA), and luminal B (LumB) sub-groups; (**B**) ROC curve analysis of miR-194-5p expression, sorted by area under the curve (AUC) in TCGA-BRCA; and (**C**) miRNA-194-5p down-regulation in the overall metastasis progression analysis by RNA sequencing.

**Figure 3 ijms-23-00325-f003:**
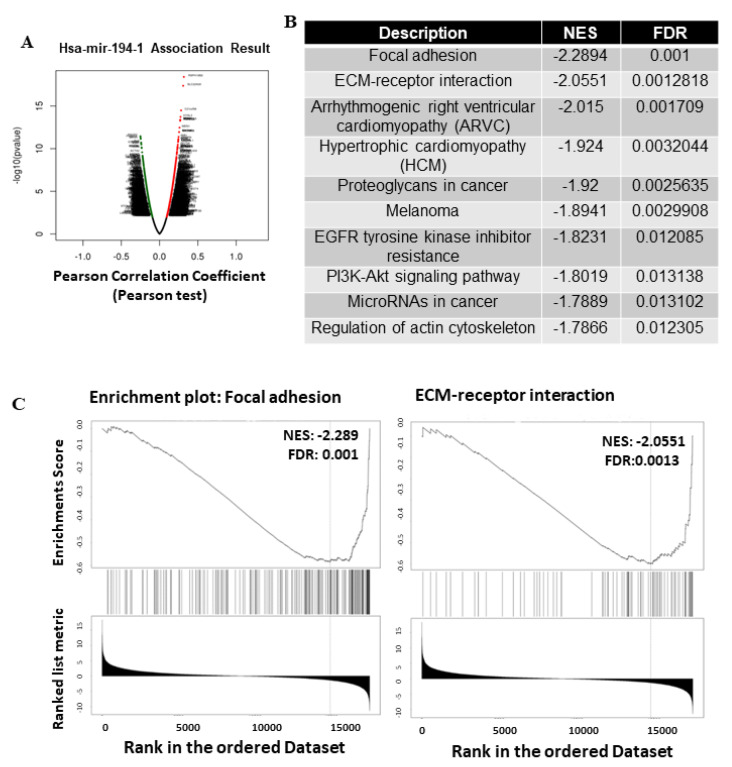
miR-194-5p is associated with disease aggressiveness in breast cancer. Key genes targeted by miR-194-5p were analyzed GSEA and GO enrichment assay in breast cancer: (**A**) The X-axis represents the log_2_(fold change), and the volcano plot shows the log_2_(fold change) vs. −log_10_(*p*-value); (**B**) negatively-regulated genes by miR-194-5p were categorized. Top 10 pathways selected from KEGG; (**C**) top two enrichment plots from the GSEA results, including focal adhesion and ECM-receptor interaction. GSEA revealed negative enrichment of miR-194-5p-altered genes in focal adhesion and ECM–receptor interactions.

**Figure 4 ijms-23-00325-f004:**
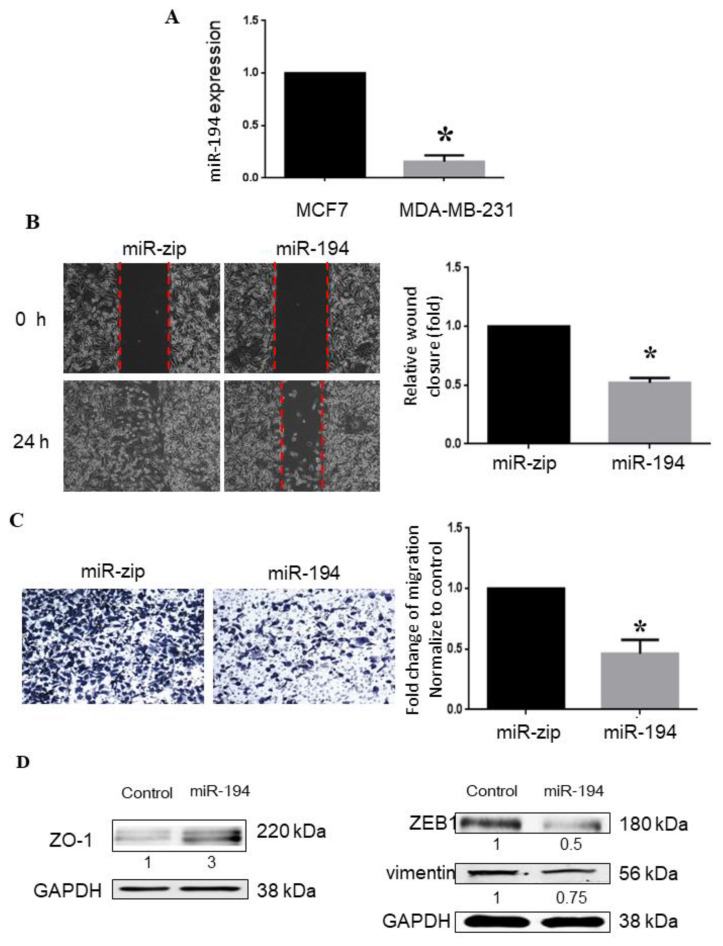
miRNA-194-5p inhibits the migration of breast cancer cells: (**A**) The expression levels of miR-194-5p were assessed by RT-qPCR in metastatic MDA-MB-231 cells and MCF7 cells. The relative miR-194-5p expression was normalized to U6 expression. (**B**) Wound-healing assay was used to evaluate whether miR-194-5p inhibited cell migration ability. MDA-MB-231 cells were observed at the indicated time points. Images of the wound gap were photographed at 0 and 24 h (at 50× magnification). Red dotted lines represented the baseline. (**C**) Cell migration abilities were evaluated by Trans-well migration assays in MDA-MB-231 cells (at 200× magnification). *, *p* < 0.05; and (**D**) the epithelial marker ZO-1 and the mesenchymal markers ZEB1 and vimentin were detected by Western blotting in transfected cells. ZO-1 protein levels were increased, while ZEB1 and vimentin protein levels were decreased in miR-194-5p transfected cells, compared with the empty vector group. miR-194-5p modulated the expression of EMT markers in MDA-MB-231 cells. The results are expressed as the mean ± SD (n = 3).

**Figure 5 ijms-23-00325-f005:**
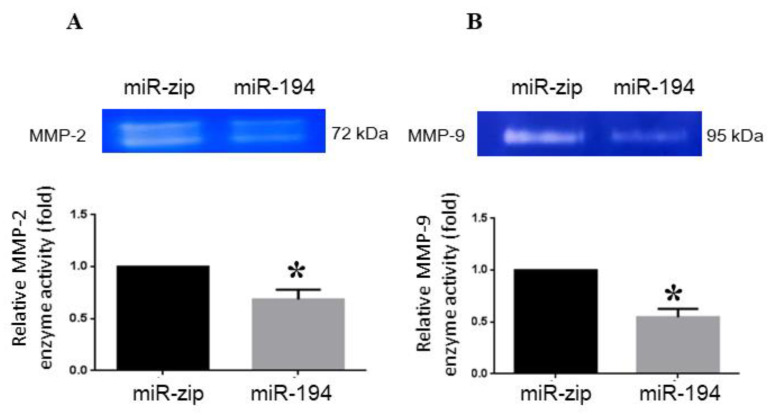
Effect of miR-194-5p on the enzyme activities of MMP-2 and MMP-9 in MDA-MB-231 cells: MDA-MB-231 cells were transfected with miR-194-5p precursors and control microRNA for 48 h, and the enzyme activities of MMP-2 (**A**) and MMP-9 (**B**) in conditioned media were analyzed by gelatin zymography. *, *p* < 0.05.

**Figure 6 ijms-23-00325-f006:**
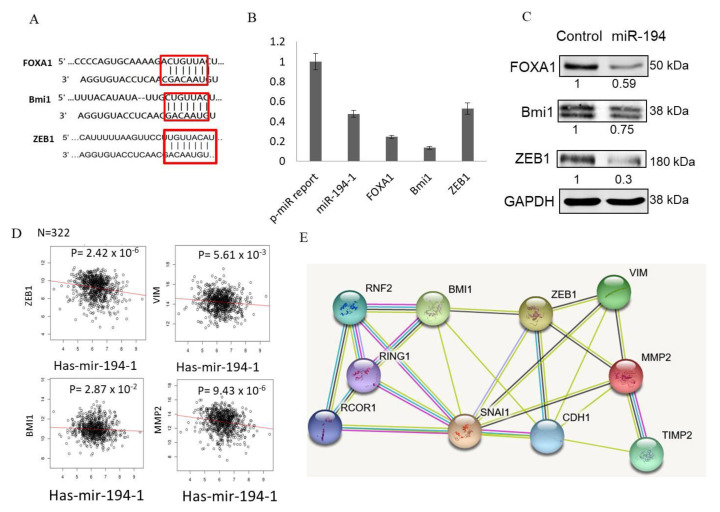
Negative correlation of miR-194-5p and EMT-associated genes in breast cancer cells. The target relationships between miR-194-5p and FOXA1, Bmi1, and ZEB1. (**A**) The target genes were predicted by the TargetScan database. The seed sequence of miR-194-5p and target binding sites were shown in the red box. (**B**) Luciferase activity was significantly decreased following co-transfection with pMIR-REPORT-3′UTR- FOXA1, Bmi1, ZEB1, and miR-194-5p in MDA-MB-231 cells. (**C**) Over-expression of miR-194-5p may down-regulate the protein expression levels of FOXA1, Bmi1, and ZEB1 in MDA-MB-231 cells. (**D**) Graphs showing the levels of miR-194-5p versus ZEB1, VIM, and MMP2 in 322 tumors from the TCGA breast cancer cohort. *p*-values were calculated using Pearson correlation tests. RPKM: Reads per kilobase per million mapped reads. (**E**) STRING-DB analysis shows that differentially expressed proteins participate in metastasis. ZEB1 and BMI1 are two major hubs in the network; colored nodes indicate query proteins; lines connected with colored nodes represent an interaction between two query proteins; different colored lines indicate information from different resources; blue lines denote data analyses from curated databases; pink lines denote data analyses from determined experiments; yellow lines denote data analyses from text mining; and black lines represent pairs of co-expressed proteins.

**Figure 7 ijms-23-00325-f007:**
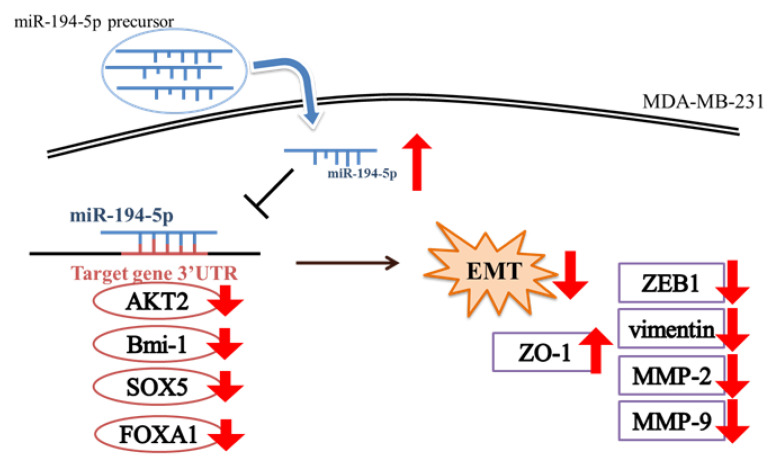
Schematic diagram of miR-194-5p regulating cell migration by ZEB1 in MDA-MB-231 cells.

## Data Availability

The original contributions presented in the study are included in the article/Appendix A. Further inquiries can be directed to the corresponding authors.

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
