# Peer review of "Down-Regulation of miR-194-5p for Predicting Metastasis in Breast Cancer Cells"

_ijms, 2021, doi:10.3390/ijms23010325_

Round 1
Reviewer 1 Report
The Authors, in their work, analysed the role of miR-194-5p in the regulation of metastasis in breast cancer cells. The topic of the manuscript is attractive; however, many points need clarification. Therefore, this manuscript cannot be published in this form.
- The name miR-194 is used throughout the text of the manuscript. The authors should specify whether it is miR-194-5p or miR-194-3p.
- Figure 1B: there is no description for this figure in the figure 1 legend, and no reference in the manuscript text.
- There are no units in figures: 2, 4 and 5.
- Figure 2C: This is a wrong graph, other than that described in the figure legend.
- Figure 4B: Does the graph actually show the relative migrated cells or the width of the wound? The authors should explain this issue.
- Figure 6: The authors should perform control experiments with mutated miRNA binding sites as well as with an empty pMIR-REPORT vector.
- Results in section 2.9 are preliminary, based on only bioinformatic analysis without experimental verification, therefore should be removed or deepened.
- Material and methods section:
- The Authors should provide the sequences of primers used for cloning of miR-195-5p binding sites in FOXA1, BMI1 and ZEB1 transcripts in pMIR-REPORT vector.
- The discussion in this publication is too general. The Authors should discuss their results in the context of other published work instead of just repeating the description of their results.
- Manny grammar and spelling mistakes, such as fragmentary sentences in lines:119-120; 146; 224-226.
Author Response
Reviewer 1:
The Authors, in their work, analysed the role of miR-194-5p in the regulation of metastasis in breast cancer cells. The topic of the manuscript is attractive; however, many points need clarification. Therefore, this manuscript cannot be published in this form.
- The name miR-194 is used throughout the text of the manuscript. The authors should specify whether it is miR-194-5p or miR-194-3p.
Response: Thank you very much for your great comments. In the revised manuscript, we specified miR-194 for miR-194-5p (miR-194-1).
- Figure 1B: there is no description for this figure in the figure 1 legend, and no reference in the manuscript text.
Response: Thank you very much for your great comments. In the revised version, we added “A forest plot displaying the number of tumor and normal samples, the area under the ROC curve (AUC), and 95% confidence interval (CI) of the AUC for each cancer type in TCGA is used to visualize the result of pan-cancer ROC analysis. The forest plot is also generated to visualize the pan-cancer Kaplan Meier (KM) survival analysis by showing the number of tumor samples, hazard ratio (HR), 95% confidence interval (CI) of the HR, and P value for each TCGA project by an CancerMIRNome bioinformatic analysis (Figure 1B) [23]. ” in manuscript text. Please refer to in line 77-84.
In addition, we added “(B) A forest plot visualizing pan-cancer survival analysis across all TCGA projects.” in Figure 1 legend. Please refer to in line 88-89.
- There are no units in figures: 2, 4 and 5.
Response: Thank you very much for your great comments. In the revised manuscript, we added “reads per million” in Figure 2A and “miRNA read counts” in Figure 2C.
We changed “Relative migrated cells” into “Relative wound closure (fold)” in Figure 4B and changed “Relative migrated cells” into “Fold change of migration-Normalize to control” in Figure 4C.
In addition, we changed “MMP-2 activity” into “Relative MMP-2 enzyme activity (fold)” in Figure 5A and changed “MMP-9 activity” into “Relative MMP-9 enzyme activity (fold)” in Figure 5B.
- Figure 2C: This is a wrong graph, other than that described in the figure legend.
Response: Thank you very much for your great comments. In the revised version, we added “Once aberrantly expressed miRNAs were identified, we further wanted to verify miRNA-194-5p for the development of Breast cancer metastases in mouse model [24]. miRNA RNA sequence analysis was performed, followed by analysis of the number of miRNAs whose expression was altered in comparison with the corresponding time point control. The downregulation of miR-194-5p observed in study at days 3 and 10 post injection of 4T1 cells (Figure 2C).” in manuscript text. Please refer to in line 104-109.
In addition, we added “(C) The miRNA-194-5p down-regulated in the overall metastasis progression analysis by RNA sequencing.” in Figure 2C legend. Please refer to in line 115-116.
- Figure 4B: Does the graph actually show the relative migrated cells or the width of the wound? The authors should explain this issue.
Response: Thank you very much for your suggestion. To avoid the vague meanings, we changed “Relative migrated cells” into “Relative wound closure (fold)” in Figure 4B
- Figure 6: The authors should perform control experiments with mutated miRNA binding sites as well as with an empty pMIR-REPORT vector.
Response: Thank you very much for your suggestion. We have performed an empty pMIR-REPORT vector as control in Figure 6A and 6B. To demonstrate the specific binding, we did western blotting analysis for miR-194-5p target proteins in Figure 6C. Furthermore, we newly added the BMI1 data in Figure 6D to show the direct target relationship between miR-194-5p and miR-194-5p target proteins.
- Results in section 2.9 are preliminary, based on only bioinformatic analysis without experimental verification, therefore should be removed or deepened.
Response: Thank you very much for your suggestion. We have modified the statement as the following table.
|
Modified version |
Original version |
|
A working model of miR-194-5p regulating cell migration by downregulating the epithelial marker ZO-1 and upregulating the mesenchymal markers ZEB1 and vimentin in MDA-MB-231 cells (Figure 7). |
2.9. Analysis of Super-Enhancer-Like Regions Verifies a Negative Correlation Between MiRNA-194 and ZEB1 Super-enhancers function as a chromatin sensor of signaling and drive miRNAs or genes to produce for tumorigenesis [25]. ROSE (Rank Ordering of Super-Enhancers) algorithm was used to catalog super-enhancer regions for different cell and tissue types based on the histone H3 lysine 27 acetylation (H3K27ac) ChIP-seq signals [26]. In this study, we carried out clustering regulatory regions analysis of miRNA-194 and ZEB1 using the 64 H3K27ac ChIP-seq data sets across 15 human tissues from the Encyclopedia of DNA Elements (ENCODE). Data showed that significant enrichment of H3K27ac signals surrounding miRNA-194 and ZEB1 are reversed, which suggests a negative correlation with an expression between miRNA-194 and ZEB1 (Figure 7). A working model of miR-194 regulating cell migration by downregulating the epithelial marker ZO-1 and upregulating the mesenchymal markers ZEB1 and vimentin in MDA-MB-231 cells (Figure 8). |
- Material and methods section:
- The Authors should provide the sequences of primers used for cloning of miR-195-5p binding sites in FOXA1, BMI1 and ZEB1 transcripts in pMIR-REPORT vector.
Response: Thank you very much for your suggestion. In the revised version, we added the following primer sequences information at the material and methods section in manuscript text.
“The following primer sequences were used for cloning miR-194-5p and miR-194-5p target genes: miR-194-5p forward, 5’-CTAGTTCCACATGGAGTTGCTGTTACAGGATCCA -3’ and reverse, 5’-AGCTTGGATCCTGTAACAGCAACTCCATGTGGAA-3’; FOXA1 forward, 5’- CTAGTCCCCAGTGCAAAAGACTGTTACTGGATCCA -3’ and reverse, 5’-AGCTTGGATCCAGTAACAGTCTTTTGCACTGGGGA -3’; Bmi1 forward, 5’-CTAGTTTTACATATATTGCTGTTACTGGATCCA-3’ and reverse, 5’-AGCTTGGATCCAGTAACAGCAATATATGTAAAA -3’; ZEB1 forward, 5’- CTAGTCATTTTTAAGTTCCTTGTTACATGGATCCA -3’ and reverse, 5’- AGCTTGGATCCTGTAACAGCAACTCCATGTGGAA -3’ ”
- The discussion in this publication is too general. The Authors should discuss their results in the context of other published work instead of just repeating the description of their results.
Response: Thank you very much for your suggestion. We have largely reorganized the Introduction and Discussion section.
- Manny grammar and spelling mistakes, such as fragmentary sentences in lines:119-120; 146; 224-226.
Response: Thank you very much for your suggestion. We have modified the statement as the following table.
|
Modified version |
Original version |
|
lines:128-129: GSEA revealed negative enrichment of miR-194-altered genes in top two gene sets including focal adhesion and ECM receptor interaction (Figure 3C). |
lines:119-120: Gene sets including focal adhesion and ECM receptor interaction (Figure 3C).
|
|
lines:155-156: ZO-1 was significantly downregulated by miR-194-5p while ZEB-1 and vimentin were significantly upregulated (Figure 4D). |
lines:146: MiR-194 significantly down-regulated ZO-1 and up-regulated ZEB-1 and vimentin (Figure 4D). |
|
lines: 223-224: Figure 7 shows how miR-194 regulates cell migration by downregulating epithelial marker ZO-1 and upregulating mesenchymal markers ZEB1 and vimentin in MDA-MB-231 cells. |
lines:224-226: A working model of miR-194 regulating cell migration by downregulating the epithelial marker ZO-1 and upregulating the mesenchymal markers ZEB1 and vimentin in MDA-MB-231 cells (Figure 8). |

Reviewer 2 Report
Overall I think this is an interesting study and well written and it should be published.
Please consider the following points to improve the manuscript:
- There are quite a few grammatical errors throughout the text, particularly in the Introduction and Discussion section which do not quite make sense. Please check the whole article and amend as appropriate.
- Abstract line 15 - there is a full stop in the wrong place
- Introduction line 34, please give the number in digits (250,000) - this sentence also does not make grammatical sense and needs re-writing
- introduction line 35-36 - this begins the discussion regarding treatment modality. I think it would make sense to begin this section with an overarching statement to include whether you are talking about primary or metastatic breast cancer - I assume we are talking about the latter
- line 41 I would say 'is not effective' instead of 'can not be used'
- line 47-49 this sentence is not grammatically correct and needs re-writing
- Line 67 -69 is this sentence needed? This is the first time you have talked about radiation and ATM-CHK-1, considering removing this sentence
- Discussion line 235 - what do you mean by 'breast cancer patients do not recover well' ?
- End of discussion section - can you expand on what we should do next based on your findings?
Author Response

(The authors gave the same response as above.)

Round 2
Reviewer 1 Report
I have no further objections to the publication of the manuscript. The authors responded to all my comments.